# Integer Scale: A Free Lunch for Faster Fine-Grained Quantization of LLMs

## Abstract

We introduce *Integer Scale*, a novel post-training quantization scheme for large language models that effectively resolves the inference bottleneck in current fine-grained quantization (*i.e.* group-wise quantization) approaches while maintaining similar accuracies. Integer Scale is a free lunch as it requires no extra calibration or fine-tuning which will otherwise incur additional costs. It can be used plug-and-play for most fine-grained quantization methods. Its integration results in at most **1.85×** end-to-end speed boost over the original counterpart with comparable accuracy. Additionally, due to the orchestration of the proposed Integer Scale and fine-grained quantization, we resolved the quantization difficulty for Mixtral-8x7B and LLaMA-3 models with negligible performance degradation, and it comes with an end-to-end speed boost of **2.13×**, and **2.31×** compared with their FP16 versions respectively.

## 1 Introduction

The size of language models has continued to grow exponentially throughout recent years. To name some iconic models, Transformers (Vaswani et al., 2017) initially bear 65M parameters, BERT (Devlin et al., 2019) exceeds with 340M, GPT-3 (Brown et al., 2020) prevails with 175B, PaLM (Chowdhery et al., 2022) trumps with 540B and most lately GPT-4 (OpenAI, 2023) is estimated to have reached 1.8T parameters. This seemingly unstoppable trend is largely promoted by the so-called scaling law (Kaplan et al., 2020) where a model's capability, via a proxy metric of auto-regressive maximum-likelihood loss, exhibits a power-law relationship to its number of parameters, dataset sizes, and compute respectively. Not surprisingly, the intimidating number of parameters of Large Language Models (LLMs) place an almost insurmountable hurdle for inference, potentially preventing their pervasive applications.

However, optimizing the serving efficiency of LLMs is a non-trivial task. LLMs generally comprise a compute-intense *pre-filling* stage and a memory-bound *self-decoding* stage. Exploiting integer matrix multiplication speeds up the computation, but directly applying post-training quantization usually generates a large performance drop. Quantization-aware training methods like LLM-QAT (Liu et al., 2023) require costly computing resources to fine-tune all the weights. In contrast, post-training quantization is more affordable and commonly used in practice. For instance, SmoothQuant (Xiao et al., 2023) transforms activation outliers into weights for better quantization accuracy. Recently, fine-granularity grouping (*i.e.* group-wise quantization as opposed to channel-wise quantization in 'coarse-grained' quantization) (Park et al., 2022) is often used as a general paradigm to reduce the quantization errors, as in ZeroQuant (Yao et al., 2022), GPTQ (Frantar et al., 2022), AWQ (Lin et al., 2023) and FPTQ (Li et al., 2023b). FPTQ proposes a fine-grained W4A8 strategy to address the memory-bound issue as a trade-off between W4A16 and W8A8. While its high quantization accuracy benefits from fine-grained quantization, the actual inference is also stalled by inefficient operations introduced by its intrinsic computational complexity due to fine granularity.

In this paper, we are driven to design a faster fine-grained quantization scheme called *Integer Scale* that renders fewer quantization errors (Table 3) and simultaneously achieves boosted speed (see Figure 1). Our contributions are multi-fold:

1. We unveil the intrinsic inference bottleneck of fine-grained LLM quantization approaches and find a hassle-free cure, called Integer Scale, with negligible accuracy loss. Our approach

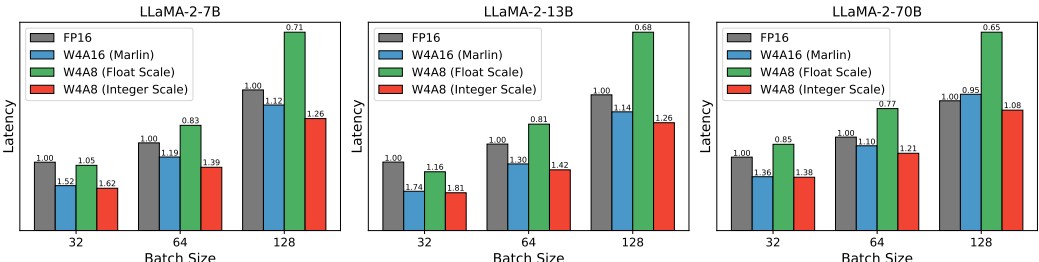

Figure 1: End-to-end latency comparison of W4A8 (Integer Scale) compared with W4A8 (Float Scale) and W4A16 (Marlin) on LLaMA-2 models. The speedup ratio is written on top of the bars.

can be used as an out-of-box plugin for the state-of-the-art quantization methods (e.g. GPTQ (Frantar et al., 2022), AWQ (Lin et al., 2023), Omniquant (Shao et al., 2023), QuaRot (Ashkboos et al., 2024) etc.) with minimum modifications.

2. The orchestration of fine-grained quantization and the integer scale scheme not only retains the performance of the existing methods but also effectively addresses the quantization difficulty of LLMs built with the mixture-of-experts technique (Jiang et al., 2024) and LLaMA-3 (AI@Meta, 2024).

3. Our integer scale, when applied to fine-grained W4A8 paradigms, achieves at most **1.85×** end-to-end speed boost over FP16, **1.17×** over Marlin W4A16 (Frantar & Alistarh, 2024), **1.83×** over its float scale counterpart, while being comparable in performance. This suggests the viability of our approach as we have achieved a new Pareto-front of speed vs. accuracy.

## 2 RELATED WORK

### 2.1 LLM SERVING FRAMEWORKS AND OPTIMIZATION TECHNIQUES

vLLM (Kwon et al., 2023) brings about paged attention (Kwon et al., 2023) and continuous batching. FasterTransformer (NVIDIA, 2023b) provides a highly optimized inference framework featuring cutlass GEMMs, CUDA kernels. Built on top of FasterTransformer (NVIDIA, 2023b), LMDeploy (Contributors, 2023) features an efficient backend called TurboMind that seeks extreme optimization through persistent batching, KV caching, and a low-bit quantization toolkit. Another sprout from FasterTransformer is TensorRT-LLM (NVIDIA, 2023c), which is tailored particularly for NVIDIA GPUs and ensembles many up-to-date inference techniques like flash attention (Dao et al., 2022), FP8 quantization (Micikevicius et al., 2022), in-flight batching, graph optimization, etc. Marlin (Frantar & Alistarh, 2024) ships so far the fastest W4A16 kernel along with a bag of optimization tricks, while QServe (Lin et al., 2024) brings an advanced W4A8 kernel implementation. FP6-LLM (Xia et al., 2024) delicately devises a software solution to support the FP6 precision on NVIDIA A100 GPUs.

### 2.2 LLM QUANTIZATION ALGORITHMS

Quantization is one of the most adopted optimization techniques to compress LLMs to their extremity. Nevertheless, it becomes more challenging as we chase for the quantization of lower bit widths (*e.g.* 4-bit, 2-bit, or binary), it faces more critical accuracy loss. It also requires efficient hardware-aware implementations that demand strenuous engineering effort.

**Weight-only Quantization.** GPTQ (Frantar et al., 2022) renovates OBQ (Frantar & Alistarh, 2022) to obtain an approximate second-order method that compensates for the quantization error. AWQ (Lin et al., 2023) is a mixed-precision weight-only method that locates salient weight channels and searches for the corresponding optimal scales. Omniquant (Shao et al., 2023) introduces learnable weight clipping that restricts extreme weight values and proposes learnable smoothing factors that tackle the activation outliers following SmoothQuant (Xiao et al., 2023). Extreme low-bit approaches also focus on weight-only quantization. Norm Tweaking (Li et al., 2024a) exploits layer norm tuning to alleviate the performance degradation, QuiP (Chee et al., 2024) profits from orthogonal matrices and

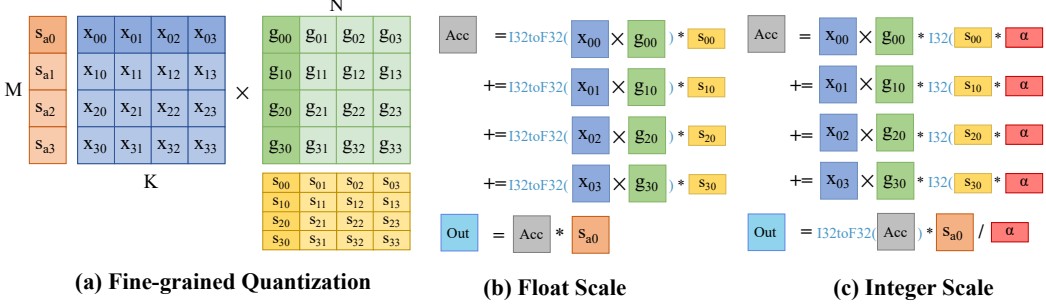

**(a) Fine-grained Quantization**    **(b) Float Scale**    **(c) Integer Scale**

Figure 2: (a) Fine-grained quantization divides activation $X$ of size $M \times K$ and weight $K \times N$ into groups for separate quantization. (b) The previous float scale scheme requires numerous costly type conversions (I32toF32) from grouped matrix multiplication results, which impedes the overall performance. Our proposed scheme (c) with integer scales and automatic amplifiers (denoted as $\alpha$) alleviates the problem while retaining similar accuracy. Note $s_{ij}$ are the scales for each weight group $g_{ij}$, and $s_{ai}$ are the scales for $X$.

AQLM (Egiazarian et al., 2024) from additive quantization with a codebook for 2-bit quantization, while PB-LLM (Shang et al., 2023) uses partial 1-bit quantization.

The weight-only scheme alleviates the memory-bound issue but its activation remains in FP16. Recent speculative parallel decoding methods (Leviathan et al., 2023; Li et al., 2024b; Cai et al., 2024) lead the decoding phase to a compute-bound scenario, which leaves room for improvement.

**Weight-Activation Quantization.** ZeroQuant (Yao et al., 2022) presents a fine-grained quantization scheme coupled with distillation. SmoothQuant (Xiao et al., 2023) enables W8A8 post-training quantization by smoothing the outliers with a heuristic factor and ships with a handcrafted CUDA kernel that ensures hardware efficiency. OdysseyLLM (Li et al., 2023a) is a coarse-grained W4A8 scheme that reduces the performance gap compared with W4A16 and W8A8. QUIK (Ashkboos et al., 2023) implements W4A4 quantization with mixed-precision.

Fine granularity generally further enhances the quantized accuracy. FPTQ (Li et al., 2023b) is a W4A8 fine-grained solution. Atom (Zhao et al., 2023) is a fine-grained mixed-precision W4A4 method. However, they typically suffer from low latency issues which cancel out the benefits from lower bit widths. DGQ (Zhang et al., 2024) attempts to apply a dual quantization scheme to improve the efficiency of the fine-grained approach.

**Rotation-based Quantization.** QuiP (Chee et al., 2024), QuiP# (Tseng et al., 2024), QuaRot (Ashkboos et al., 2024) are a line of quantization methods that profits from the computation invariance of the orthogonal matrices for outlier suppression. To undo the rotation effect, extra online transformations are applied. When implemented efficiently, this overhead can be deemed nearly negligible.

## 3 MOTIVATION

### 3.1 FINE GRANULARITY STRENGTHENS CURRENT QUANTIZATION APPROACHES

Fine granularity approaches (Li et al., 2023b; Lin et al., 2023; Zhao et al., 2023) bear prevailing benefits over many state-of-the-art LLM quantization methods. In extreme cases, it even produces reasonable results when coarse methods fail. It can be applied as a plug-in method to boost the accuracy of the existing methods. Formally, the output $\mathbf{O}_i$ of a fine-grained weight-activation quantization GEMM can be written as,

$$\mathbf{O}_i = \mathbf{s}_{a_i} * \sum_g (\mathbf{X}_{g_i} \times \mathbf{W}_{g_i}^\top) * \mathbf{s}_{g_i} \tag{1}$$

where $\mathbf{s}_{a_i}$ is the $i$-th scale for the activation, $\mathbf{s}_{g_i}$ is the scale for each weight group. $\mathbf{X}_{g_i}$ and $\mathbf{W}_{g_i}$ are the corresponding activation and weight for each group $g$. Depending on the precision of matrix

multiplication, specific type conversions are required to perform either scalar or matrix multiplication. For instance, if we adopt a fine-grained W8A8 scheme with integer tensor cores for the computation, the INT32 result has to be converted to float for the later dequantization.

This process is depicted in Figure 2 (a), where it typically considers weights in groups and each has its float scale. We apply the fine-granularity strategy to approaches that cover commonly-used bit widths range in W4A16, W8A8, W4A8, and W4A4 in Table 1 to exhibit that group-wise fine-granularity consistently improves the quantized performance compared with its original coarse counterpart. Note on the LLaMA-3-70B model, the vanilla Round-to-Nearest (RTN) caused a large performance collapse while its fine-grained version can easily handle it. As we drive from W8A8 to lower bits, the quantization error increases. Especially, when applying QuaRot (Ashkboos et al., 2024) on LLaMA-3-70B at W4A4, the perplexity bursts into an unreasonable value, and fine-granularity can alleviate the issue.

Table 1: Applying fine granularity (denoted by 'FG') to the state-of-the-art quantization methods on LLaMA-2 models. Perplexity is tested on C4 (the lower the better). Group = -1 indicates coarse-grained weight quantization while 128 means fine-grained with a group size of 128.

| Bitwidth | Method | Group | LLaMA-2 | | | LLaMA-3 | |
|---|---|---|---|---|---|---|---|
| | | | 7B | 13B | 70B | 8B | 70B |
| FP16 | Baseline | | 7.05 | 6.46 | 5.52 | 8.88 | 6.73 |
| W8A8 | RTN (Yao et al., 2022) | -1 | 7.19 | 6.51 | 5.64 | 9.05 | 75.05 |
| | RTN w/ FG | 128 | 7.2 | 6.51 | 5.64 | 9.04 | 7.15 |
| W8A8 | SmoothQuant (Xiao et al., 2023) | -1 | 7.2 | 6.51 | 5.58 | 9.03 | 7.38 |
| | SmoothQuant w/ FG | 128 | 7.2 | 6.51 | 5.58 | 9.03 | 7.48 |
| W8A8 | FPTQ (Li et al., 2023b) | -1 | 7.08 | 6.50 | 5.55 | 8.97 | 8.88 |
| | FPTQ w/ FG | 128 | 7.08 | 6.50 | 5.54 | 8.95 | 6.81 |
| W4A16 | GPTQ (Frantar et al., 2022) | -1 | 7.47 | 6.84 | 5.71 | 10.54 | 7.83 |
| | GPTQ w/ FG | 128 | 7.22 | 6.65 | 5.61 | 9.70 | 7.26 |
| W4A8 | Odyssey (Li et al., 2023a) | -1 | 7.58 | 6.70 | 5.78 | 10.25 | 12.15 |
| | Odyssey w/ FG | 128 | 7.26 | 6.60 | 5.60 | 9.56 | 7.09 |
| W4A4 | QuaRot (Ashkboos et al., 2024) | -1 | 7.87 | 7.11 | 5.92 | 12.06 | 544.50 |
| | QuaRot w/ FG | 128 | 7.82 | 7.08 | 5.90 | 11.8 | 132.20 |

## 3.2 FINE-GRAINED QUANTIZATION SUFFERS FROM THE INFERENCE BOTTLENECK

Although fine-grained quantization can achieve higher accuracy, as demonstrated in (Li et al., 2023b), we have found it to be particularly slow during inference, which is also noted in the Dual-Granularity Quantization (DGQ) method (Zhang et al., 2024). The advantages of using lower bit widths are often offset by the computational overhead they introduce. Figure 3 compares the kernel latency under typical inference batch sizes (drops from 3.15× to 0.5×). Notably, the fine-grained kernel is significantly slower when compared to FP16 at larger batch sizes, making it less practical for deployment. Further analysis confirms that fine-grained approaches inherently require numerous costly type conversions. The result of each integer matrix multiplication has to be converted to float precision to multiply the corresponding float scale, as depicted in Figure 2 (b).

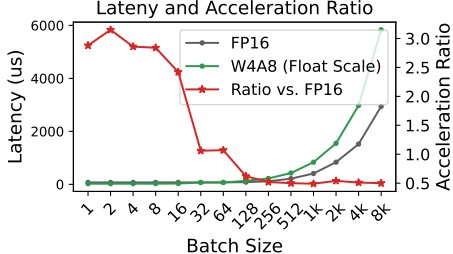

Figure 3: Kernel latency comparison between W4A8 w/ Float Scale vs. FP16. The red line denotes its acceleration ratios over FP16.

The intrinsic computation drawbacks disable its use in practice. This incoherent situation calls for a novel fine-grained scheme that is both computationally efficient and accuracy-retaining.

## 4 METHOD

### 4.1 INTEGER SCALE WITH ADAPTIVE SCALE AMPLIFIER

Motivated by the previous discussion, it is then critical to boost the fine-grained inference. Figure 2 (b) has shown that using float scale triggers numerous costly type conversions. For instance, a typical `Dense` layer of size 4096×4096 with 128 groups has 131072 float scales, thus the same amount of type conversion operations are needed. Each operation requires additional element-wise conversions. Intuitively, we can resort to integer scales to avoid it. However, since all normalized float scales are in the $(0, 1)$ range, directly converting scales to integers certainly causes tremendous quantization errors. To mitigate this problem, we involve an integer amplifier $\alpha$, called *adaptive scale amplifier*, which can be easily computed based on the available float scales. Our method is put formally as,

$$\mathbf{O}_i = \mathbf{s}_{a_i} * \text{FLOAT}\big(\sum_g (\mathbf{X}_{g_i} \times \mathbf{W}_{g_i}^\top) * \text{INT}(\mathbf{s}_{g_i} * \alpha)\big)/\alpha \tag{2}$$

To find the common amplifier, we use a heuristic search algorithm that starts from $2^0$ to amplify the minimum scale of all groups until we meet an amplifier $2^i$ that guarantees amplified scales to be bigger than 1, see Listing 1. Note we adopt an amplifier as the power of 2 for more efficient implementation of multiplication and division as simple bit shifts will do. It is not necessarily to be so, other amplifiers like $INT(1/tmp)$ will also be fine.

Ideally, we can use the above heuristic method to find the optimal amplifier per layer. However, based on the scale analysis of LLaMA-2-7B in Figure 4 (a,b,c), we find that the number of bit shifts required to amplify the scale mainly falls to 9 or 10. The weight MSE when using an amplifier of $2^{10}$ is in the range of $(10^{-7}, 10^{-6})$, as compared with its float counterpart. A similar observation applies to 13B and 70B models. We can select $2^{10}$ as our default amplifier to avoid

```
1 scale_min = scales.min()
2 n, tmp = 0, scale_min
3 while tmp < 1:
4     tmp = scale_min * (2**n)
5     n+=1
6 scale_amplifier = 2**(n-1)
```
Listing 1: Quick Heuristic Search for Integer Scale Amplifier

possible overflow as the later ablation (Table 8) shows a bigger amplifier has no clear gains.

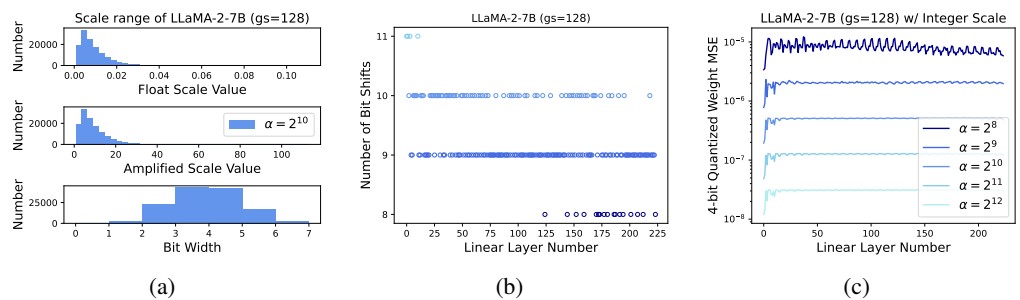

(a)  (b)  (c)

Figure 4: (a) The range of amplified ($\alpha = 2^{10}$) float scales of LLaMA-2-7B in the first layer (others are similar) mapped to 16-bit integers. The majority of amplified scales can be represented within 8 bits. (b) The number of bit shifts required to amplify scales per linear layer. (c) Weight MSE between integer scale and float scale under different amplifiers.

### 4.2 KERNEL IMPLEMENTATION

Table 2 illustrates the difference in typical kernels. Current hardware supports a standard `MatMul` GEMM which isn't suited for fine-grained approaches. Each group of $A_i$ and $W_i$ are multiplied and iteratively accumulated to register $C_i$. Atom (Zhao et al., 2023)'s fine-grained W4A4 kernel adopts 4-bit for both weight and activation, which performs group-wise products and collects partial sums with an additional register $C'$. Note Atom's $float$ conversion becomes the main bottleneck while ours removes this costly operation by applying integer scales $s_i^{INT}$.

Table 2: Comparison of kernel computation logic.

| MatMul | Atom | Ours |
|---|---|---|
| $C_1 = A_1 * W_1 + C_0$ | $C_1 = A_1 * W_1, C' = float(C_1) * s_1$ | $C_1 = A_1 * W_1, C'' = C_1 * s_1^{INT}$ |
| $C_2 = A_2 * W_2 + C_1$ | $C_2 = A_2 * W_2, C' += float(C_2) * s_2$ | $C_2 = A_2 * W_2, C'' += C_2 * s_2^{INT}$ |
| $\cdots$ | $\cdots$ | $\cdots$ |

We also present our computation strategy in Figure 2 (c). Since the result of group-wise weight and activation matrix multiplication (e.g., $x_{00} \times g_{00}$, executed with integer tensor cores) becomes INT32, we only need to convert the amplified scale to INT32 offline. Each group is then accumulated to have the final result. The large number of type conversions on the matrix is thus reduced to only once for activation dequantization. Besides, we exploit the efficient weight processing and kernel fusion technique of OdysseyLLM's FastGEMM (Li et al., 2023a) for fast inference. The combination makes fine-grained kernels substantially efficient, enabling fine-grained approaches as a feasible solution.

## 5 EXPERIMENTS

### 5.1 SETUP

**Models and Datasets.** We benchmark Integer Scale and other state-of-the-art quantization methods on the well-known LLaMA-2 (Touvron et al., 2023) and LLaMA-3 (AI@Meta, 2024) models and Mixtral 8x7B (Jiang et al., 2024). Several datasets are used for evaluation, including LAMBADA (Paperno et al., 2016), C4 (Raffel et al., 2020), WikiText-2 (Merity et al., 2016), MMLU (Hendrycks et al., 2021), and a set of Common Sense QA (Talmor et al., 2019) tasks like WinoGrande (Sakaguchi et al., 2021), PIQA (Tata & Patel, 2003), HellaSwag (Zellers et al., 2019), ARC$_e$. For CommonSense QA tasks, we utilized the Language Model Evaluation Harness (Gao et al., 2021) tool.

**Inference Framework.** We adopt an end-to-end inference pipeline with cutlass (NVIDIA, 2023a) that mainly profits GPU Tensor Core execution, kernel fusion policies, and graph optimization. Unless otherwise notified, we use the same framework for fair comparisons. Note for LLaMA models with W4A16, we use Marlin (Frantar & Alistarh, 2024) for inference as it claims to be the fastest available framework. For Mixtral 8x7B, we had to use our W4A16 implementation as Marlin hasn't supported it yet. The latency is tested on a single NVIDIA A100 GPU, except for LLaMA-2-70B and Mistral 8x7B we use four such GPUs.

**Baselines.** In our experiments, we choose GPTQ (Frantar et al., 2022), AWQ (Lin et al., 2023), and Omniquant (Shao et al., 2023) as our baselines, given that they are the most prevalent fine-grained quantization schemes. Throughout the paper, we adopt per-token activation quantization, and per-channel weight quantization by default.

### 5.2 EXPERIMENT RESULT ON LAMBADA, C4, AND WIKITEXT-2

Table 3 exhibits the quantization result of LLaMA-2 and Mixtral models when applying Integer Scale (IS) to GPTQ (Frantar et al., 2022), AWQ (Lin et al., 2023), and Omniquant (Shao et al., 2023) on LAMBADA, WikiText-2, and C4 datasets. Our approach generally shows on-par or better performance, indicating that the Integer Scale applies to the existing quantization methods and retains the quantized performance at lower bits like W4A8. Note since Ominiquant on LLaMA-2-70B originally fails, so does its integer scale variation.

### 5.3 EXPERIMENT RESULT ON COMMON SENSE QA

Table 4 compares the Common Sense QA (Talmor et al., 2019) result of applying the Integer Scale on state-of-the-art quantization approaches. A similar conclusion to Section 5.2 can be reached. More results on MMLU (Hendrycks et al., 2021) can be found in Table 9 in Section B.

Table 3: Comparison with state-of-the-art quantization methods on LAMBADA (accuracy), C4 (PPL), and WikiText (PPL). For all models tested, we set the weight group size to 128 and apply symmetric quantization. Integer Scale (IS) with amplifier 1024 is used.

| Dataset | HyperParam | | LLaMA-2 | | | Mixtral |
|---|---|---|---|---|---|---|
| | Method | BitWidth | 7B | 13B | 70B | 8x7B |
| | FP16 | W16A16 | 73.70% | 76.64% | 79.57% | 77.62% |
| | GPTQ | W4A8 | 71.65% | 75.88% | 78.54% | 73.89% |
| | GPTQ w/ IS | W4A8 | 71.66% $_{+0.01}$ | 75.39% $_{-0.48}$ | 78.67% $_{+0.13}$ | 73.93% $_{+0.03}$ |
| LAMBADA | AWQ | W4A8 | 70.15% | 75.47% | 78.48% | 76.24% |
| | AWQ w/ IS | W4A8 | 70.07% $_{-0.07}$ | 75.02% $_{-0.44}$ | 78.42% $_{-0.05}$ | 74.30% $_{-1.94}$ |
| | Omniquant | W4A8 | 71.76% | 75.98% | NaN | 76.09% |
| | Omniquant w/ IS | W4A8 | 70.91% $_{-0.85}$ | 75.60% $_{-0.36}$ | NaN | 76.01% $_{-0.07}$ |
| | FP16 | W16A16 | 5.65 | 4.95 | 3.36 | 3.93 |
| | GPTQ | W4A8 | 12.32 | 5.16 | 3.66 | 4.51 |
| | GPTQ w/ IS | W4A8 | 13.13 $_{+0.81}$ | 5.18 $_{+0.02}$ | 3.69 $_{+0.03}$ | 4.59 $_{+0.08}$ |
| WikiText-2 | AWQ | W4A8 | 6.12 | 5.27 | 3.66 | 4.30 |
| | AWQ w/ IS | W4A8 | 6.19 $_{+0.07}$ | 5.30 $_{+0.03}$ | 3.70 $_{+0.04}$ | 4.42 $_{+0.12}$ |
| | Omniquant | W4A8 | 5.94 | 5.16 | NaN | 4.27 |
| | Omniquant w/ IS | W4A8 | 5.97 $_{+0.03}$ | 5.17 $_{+0.01}$ | NaN | 4.36 $_{+0.09}$ |
| | FP16 | W16A16 | 7.05 | 6.46 | 5.52 | 6.88 |
| | GPTQ | W4A8 | 39.96 | 6.66 | 5.75 | 7.31 |
| | GPTQ w/ IS | W4A8 | 37.25 $_{+2.71}$ | 6.68 $_{+0.02}$ | 5.78 $_{+0.03}$ | 7.39 $_{+0.08}$ |
| C4 | AWQ | W4A8 | 7.57 | 6.79 | 5.73 | 7.15 |
| | AWQ w/ IS | W4A8 | 7.64 $_{+0.07}$ | 6.83 $_{+0.04}$ | 5.76 $_{+0.03}$ | 7.27 $_{+0.12}$ |
| | Omniquant | W4A8 | 7.41 | 6.65 | NaN | 7.12 |
| | Omniquant w/ IS | W4A8 | 7.44 $_{+0.03}$ | 6.67 $_{+0.02}$ | NaN | 7.21 $_{+0.09}$ |

## 5.4 W4A8 Kernel Latency Comparison

Figure 5 (a) illustrates the comparison of kernel implementations under various bandwidths. Marlin (Frantar & Alistarh, 2024) ships so far the most advanced W4A16 kernel implementation. Odyssey's W4A8 scheme largely benefits its specific FastGEMM and has the optimal acceleration ratio over FP16. It can be seen that fine-grained W4A8 with integer scale becomes a feasible scheme between W4A16 and non-fine-grained W4A8 for better accuracy. Interestingly, we discover a "performance cliff" (gray-colored) where the acceleration ratio suddenly drops when it transits from memory-bound to compute-bound scenarios. This is due to the sudden drop from the ideal 4× speedup to 2× vs. FP16. It is however as expected. In small batch scenarios where the inference is mainly memory-bound, all W4 solutions could achieve nearly theoretical 4× acceleration (W4 vs. FP16). While in large batch scenarios where it is leaning towards compute-bound, the ratio turns into 2× since INT8 tensor cores are theoretically 2× of FP16 (A8 vs. FP16). However, the fine-grained W4A8 kernel with float scale suffers from fine granularity and its speed is even inferior to W4A16, for which reason we resolve it with Integer scale to achieve practical gain.

## 5.5 Speed boost on Mixture-of-experts

Figure 5 (c) shows the end-to-end latency of the W4A8 Integer Scale scheme applied to the Mixtral 8×7B, where we obtain at most **1.55×** and **1.3×** boost, compared with FP16 and W4A16 respectively.

## 5.6 Our Recipe for LLaMA-3

LLaMA-3 is difficult to quantize at lower bits compared with its predecessors, as confirmed in (Huang et al., 2024). To counter the problem, we apply QuaRot (Ashkboos et al., 2024)

Table 5: Our LLaMA-3 Integer Scale recipe.

| Model | BitWidth | $\alpha$ | Group | C4 | WikiText-2 |
|---|---|---|---|---|---|
| LLaMA-3-8B | W4A8 | - | 128 | 9.331 | 6.352 |
| LLaMA-3-8B | W4A8 | 8192 | 128 | 9.379 | 6.382 |
| LLaMA-3-70B | W4A8 | - | 128 | 7.061 | 3.280 |
| LLaMA-3-70B | W4A8 | 8192 | 128 | 7.092 | 3.312 |

Table 4: Comparison with state-of-the-art quantization methods on Common Sense QA. For all models tested, we set the weight group size to 128 and apply symmetric quantization. Integer Scale (IS) with amplifier 1024 is used.

| Model | HyperParam | | Common Sense QA | | | | |
|---|---|---|---|---|---|---|---|
| | Method | BitWidth | WinoGrande | PIQA | HellaSwag | ARC_e | Avg |
| | FP16 | W16A16 | 0.6906 | 0.7911 | 0.7598 | 0.7458 | 0.7468 |
| | GPTQ | W4A8 | 0.6819 | 0.7829 | 0.7380 | 0.6961 | 0.7247 |
| | GPTQ w/ IS | W4A8 | 0.6882 | 0.7845 | 0.7359 | 0.6932 | 0.7255 |
| LLaMA-2-7B | AWQ | W4A8 | 0.6890 | 0.7807 | 0.7418 | 0.6856 | 0.7243 |
| | AWQ w/ IS | W4A8 | 0.6803 | 0.7818 | 0.7399 | 0.6717 | 0.7184 |
| | Omniquant | W4A8 | 0.6930 | 0.7873 | 0.7427 | 0.6890 | 0.7280 |
| | Omniquant w/ IS | W4A8 | 0.6882 | 0.7818 | 0.7393 | 0.6898 | 0.7248 |
| | FP16 | W16A16 | 0.7222 | 0.8052 | 0.7938 | 0.7744 | 0.7739 |
| | GPTQ | W4A8 | 0.7080 | 0.8003 | 0.7858 | 0.7980 | 0.773 |
| | GPTQ w/ IS | W4A8 | 0.7040 | 0.8025 | 0.7854 | 0.7917 | 0.7709 |
| LLaMA-2-13B | AWQ | W4A8 | 0.7182 | 0.7976 | 0.7758 | 0.7677 | 0.7648 |
| | AWQ w/ IS | W4A8 | 0.7246 | 0.7992 | 0.7734 | 0.7668 | 0.7660 |
| | Omniquant | W4A8 | 0.7214 | 0.7992 | 0.7810 | 0.7710 | 0.7682 |
| | Omniquant w/ IS | W4A8 | 0.7127 | 0.7954 | 0.7786 | 0.7715 | 0.7646 |
| | FP16 | W16A16 | 0.7798 | 0.8275 | 0.8381 | 0.8098 | 0.8138 |
| | GPTQ | W4A8 | 0.7664 | 0.8313 | 0.8314 | 0.8131 | 0.8106 |
| LLaMA-2-70B | GPTQ w/ IS | W4A8 | 0.7585 | 0.8324 | 0.8287 | 0.8077 | 0.8068 |
| | AWQ | W4A8 | 0.7664 | 0.8194 | 0.8202 | 0.8005 | 0.8016 |
| | AWQ w/ IS | W4A8 | 0.7624 | 0.8199 | 0.8218 | 0.7929 | 0.7993 |
| | FP16 | W16A16 | 0.7648 | 0.8368 | 0.8403 | 0.835 | 0.8192 |
| | GPTQ | W4A8 | 0.7553 | 0.8161 | 0.8272 | 0.8056 | 0.8011 |
| | GPTQ w/ IS | W4A8 | 0.7427 | 0.8145 | 0.8280 | 0.7925 | 0.7944 |
| Mixtral-8x7B | AWQ | W4A8 | 0.7506 | 0.8341 | 0.8288 | 0.8228 | 0.8091 |
| | AWQ w/ IS | W4A8 | 0.7443 | 0.8286 | 0.8252 | 0.8131 | 0.8028 |
| | Omniquant | W4A8 | 0.7553 | 0.8308 | 0.8338 | 0.8165 | 0.8091 |
| | Omniquant w/ IS | W4A8 | 0.7506 | 0.8308 | 0.8337 | 0.8178 | 0.8082 |

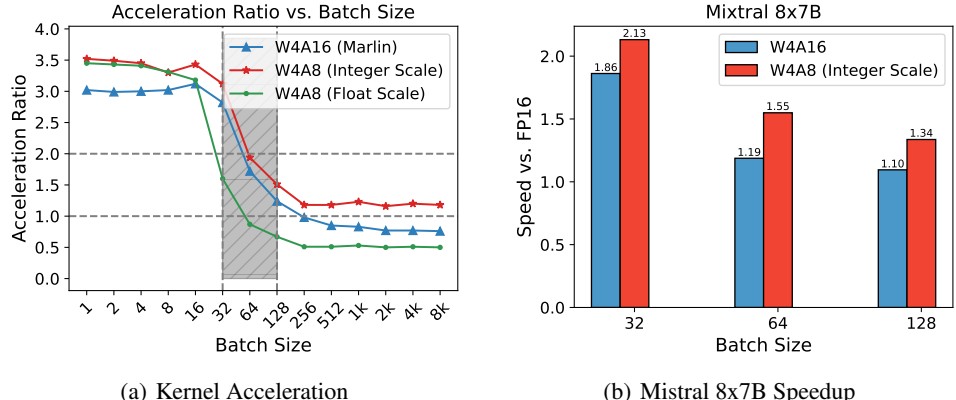

(a) Kernel Acceleration      (b) Mistral 8x7B Speedup

Figure 5: (a) Fine-grained W4A8 kernel (K=4096, N=22016) with the integer scale (W4A8 Integer Scale) boosts its float scale counterpart (W4A8 Float Scale). The gray region denotes the "performance cliff". (b) End-to-end speed boost on Mixtral 8x7B over FP16 under various batch sizes.

with a fine-grained paradigm. We adopt 8-bit per-token activation quantization and 4-bit fine-grained symmetric quantization with a group size of 128. Besides, we use fine-grained W8A8 for down projection layers following the observation in (Li et al., 2023b). Table 5 exhibits the result of our

LLaMA-3 scheme, while integer scale outperforms GPTQ's W4A16 (-1.16% in C4 perplexity) shown in Table 1.

## 5.7 COMPARISON WITH MARLIN'S W4A16 SCHEME

We compare our Integer Scale scheme with Marlin's implementation of GPTQ (Frantar & Alistarh, 2024) in Table 6. We are mostly on par with GPTQ at W4A16 when tested on C4, WikiText-2, and MMLU. Their acceleration ratios vs. FP16 are compared in Figure 5 where W4A8 surpasses W4A16 mainly due to faster tensor core execution at lower bit widths.

Table 6: C4 and WikiText-2 perplexity, and MMLU zero-shot accuracy of LLaMA-2-7B quantized with Marlin's implementation of GPTQ (W4A16) vs. GPTQ w/ Integer Scale (W4A8).

| Method | BitWidth | C4 | WikiText-2 | MMLU |
|---|---|---|---|---|
| GPTQ | W4A16 | 7.2093 | 5.8212 | 39.11% |
| GPTQ w/ Integer Scale | W4A8 | 7.4011 | 5.9433 | 38.54% |

This attests that fine-grained W4A8 with the Integer Scale is a competitive strategy in terms of both quantization loss and speed.

## 5.8 COMPARISON WITH QSERVE'S W4A8 KERNEL

Figure 6 presents the kernel speed comparison with QServe (Lin et al., 2024), which ships an advanced W4A8 kernel. For coarse-grained W4A8 kernel with M=1, K=4096, and N=22016, our W4A8 kernel execution is substantially faster than QServe at all batch sizes. A similar conclusion is affirmed for the fine-grained kernel at a typical group size of 128. Both being the same bit widths, our fine-grained kernel with Integer Scale is substantially faster than QServe's, with a maximum of being **1.53×**. It turns out that the main difference lies in the intrinsic complexity of Dual Quantization (Zhang et al., 2024) they adopted which first quantizes weights in 8-bit and again in 4-bit. Note the second step is kept asymmetric to counter quantization loss. This *asymmetric* scheme requires element-wise multiplication and subtraction that must be done in costly CUDA cores. See more details in B.2.

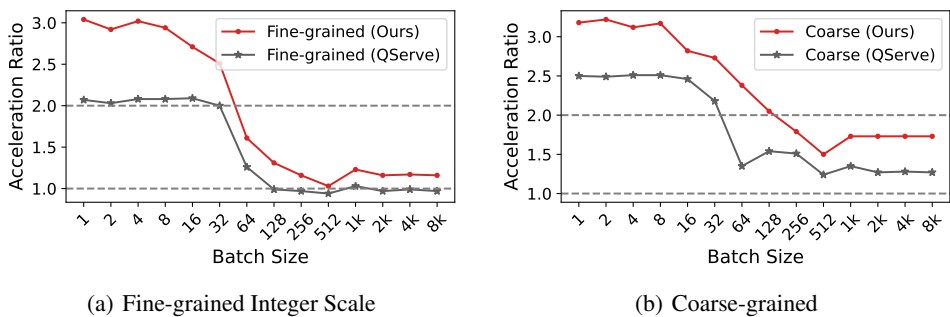

(a) Fine-grained Integer Scale        (b) Coarse-grained

Figure 6: Kernel speed comparison with QServe's W4A8 at K=4096, N=22016. The acceleration ratio is against FP16. Both our fine and coarse-grained kernels are faster.

## 5.9 COMPARISON WITH TENSORRT-LLM'S W8A8 AND W4A16KV8

Shown in Table 7, our FP16 implementation is comparable to TensorRT-LLM while our W4A8 with Integer Scale is comfortably faster than TensorRT-LLM's W8A8 under several different batch size settings (1,8,16,32). We use LLaMA-2-7B and set the input and output length to 128.

## 6 ABLATION STUDY

### 6.1 FIXED AMPLIFIER VS. HEURISTIC SEARCH

To find the optimal amplifier for the integer scale, we test several amplifiers in Table 8. It turns out that using an amplifier bigger than 1024 doesn't bring substantial gains while $2^{10}$ is a good trade-off between performance and the overflow risk. It is thus safe to amplify the scale with 1024

Table 7: End-to-end Latency Comparison (ms) with TensorRT-LLM's W8A8 on LLaMA-2-7B.

| LLaMA-2-7B | Bit Width | BS=1 | BS=8 | BS=16 | BS=32 |
|---|---|---|---|---|---|
| TRT-LLM | W8A8 | 859.76 | 1058.62 | 1171.86 | 1365.68 |
| TRT-LLM | FP16 | 1280.83 | 1411.97 | 1555.52 | 1819.03 |
| Integer Scale (Ours) | W4A8 | 533.4 | 632.37 | 831.63 | 1147.43 |
| (Ours) | FP16 | 1281.65 | 1426.87 | 1552.36 | 1802.18 |

with minimum overflow risk. To verify this choice, we draw the maximum activation per layer of LLaMA-2 and Mixtral models using $\alpha = 2^{10}$ in Figure 8 (B.4), where all values fall within $2^{31}$.

Table 8: Ablation on the amplifier value. Perplexity is tested on C4.

| BitWidth | Amplifier | LLaMA-2-7B | LLaMA-2-13B | LLaMA-2-70B | LLaMA-3-8B | LLaMA-3-70B |
|---|---|---|---|---|---|---|
| W4A16 | - | 7.43 | 6.64 | 5.66 | 10.00 | 9.06 |
| W4A16 | Heuristic | 7.46 | 6.65 | 5.66 | 10.03 | 9.10 |
| W4A16 | 128 | 6.75 | 7.57 | 5.81 | 15.52 | 13.84 |
| W4A16 | 512 | 7.45 | 6.65 | 5.67 | 10.09 | 9.27 |
| W4A16 | 1024 | 7.45 | 6.64 | 5.66 | 10.03 | 9.04 |
| W4A16 | 4096 | 7.45 | 6.64 | 5.67 | 10.00 | 8.91 |

## 6.2 SPEED COMPARISON OF FLOAT SCALE VS. INTEGER SCALE

We compare the difference in inference speed using float and integer scales to showcase the latency advantage of using the Integer scale in Figure 5 (a). The speedup is at most **2.3×**, suggesting the reduction of costly type conversions is more than necessary.

## 7 CONCLUSION

In this paper, we introduced a plug-and-play scheme called *Integer Scale* that can be applied to speed up the existing fine-grained quantization approaches. We showed through extensive experiments that the Integer Scale not only benefits from the performance boost due to fine granularity but also well resolves the intrinsic computational overhead. It can serve as a default free-lunch technique with fine-grained approaches of various bandwidths to render an overall competitive quantization strategy. Moreover, the same strategy can be applied to Mixtral 8×7B based on a mixture-of-experts and LLaMA-3, which were previously difficult to quantize at lower bit widths.

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

## A  BACKGROUND KNOWLEDGE ON LLM QUANTIZATION

### A.1  SYMMETRIC VS. ASYMMETRIC QUANTIZATION

We suggest referring to the white paper (Nagel et al., 2021) for a thorough understanding of network quantization. We draw some key concepts here as a quick manual. Both symmetric and asymmetric quantization use uniform quantization that maps float values to integer values with a single scale. Symmetric quantization computes the scale $s$ as,

$$s = \frac{|X|_{max}}{2^{n-1} - 1} \tag{3}$$

$$Q(X) = clamp(\lceil X/s \rceil, -2^{n-1}, 2^{n-1} - 1) \tag{4}$$

For asymmetric quantization, a zero point is utilized.

$$s = \frac{X_{max} - X_{min}}{2^n - 1}, z = \lceil \frac{-X_{min}}{s} \rceil \tag{5}$$

$$Q(X) = clamp(\lceil X/s \rceil + z, 0, 2^n - 1) \tag{6}$$

### A.2  PER-TENSOR, PER-TOKEN, PER-CHANNEL QUANTIZATION, GROUP-WISE QUANTIZATION

Take symmetric quantization as an example, *per-tensor quantization* uses the same scale for all tensor values. *Per-channel/token quantization* uses a scale for a row or a column of the tensor. We can divide each channel into groups for *group-wise quantization* (Lin et al., 2023), also called fined-grained quantization.

## B  ADDITIONAL DISCUSSIONS

### B.1  EXPERIMENT RESULT ON MMLU

Table 9 compares the result on MMLU (Hendrycks et al., 2021).

### B.2  MORE DISCUSSION WITH QSERVE

Due to the adopted asymmetry quantization, QServe's kernel is prone to complex computation logic that can be formulated as,

$$C_1 = A_1 * (W_1 - z_1) * s_1 + C_0 = A_1 * (W_1 * s_1 - z_1 * s_1) + C_0 \tag{7}$$
$$C_2 = A_2 * (W_2 - z_2) * s_2 + C_1 = A_1 * (W_1 * s_2 - z_2 * s_2) + C_1 \tag{8}$$
$$\cdots \tag{9}$$

where $s_i$ and $z_i$ are the $i$-th scale and zero point for dequantization. Note $W_i * s_i$ is element-wise multiplication, and the subtraction is performed with a `vadd4` instruction.

Figure 7 gives the additional comparison on kernel (N=4096, K=4096), where our fine and coarse-grained kernels also outperform QServe, indicating our flexibility in different inputs.

### B.3  COMPARISON WITH VS-QUANT AND DGQ

The contribution of our paper is to resolve the intrinsic efficiency problem that lies in fine-grained LLM quantization approaches like Atom (Zhao et al., 2023). We are different from VS-Quant (Dai et al., 2021) which was solely evaluated on ResNet-50 and BERT models. More importantly, directly quantizing scales like VS-Quant will inevitably cause clipping and rounding errors. In contrast, we use an amplifier to expand the scale to a range that is safe to convert to integers. They two are essentially different. Furthermore, VS-Quant is motivated by reducing energy overheads while we are driven by mitigating the inference bottleneck of LLMs. In Table 10, we compare two methods under similar fine-grained (group size of 128) W4A8 settings, while VS-Quant uses quantized scales with per-channel quantization (as proposed by VS-Quant in their paper). Our Integer Scale uses an

Table 9: Comparison with state-of-the-art quantization methods on MMLU. For all models tested, we set the weight group size to 128 and apply symmetric quantization. Integer Scale (IS) with amplifier 1024 is used.

| Model | HyperParam | | MMLU | | | | |
|-------|------------|----------|---------|---------|---------|---------|---------|
|       | Method | BitWidth | Hums. | STEM | Social | Other | Avg |
| LLaMA-2-7B | FP16 | W16A16 | 36.92% | 30.75% | 40.92% | 45.68% | 38.49% |
|       | GPTQ | W4A8 | 33.69% | 30.45% | 40.36% | 42.91% | 36.58% |
|       | GPTQ w/ IS | W4A8 | 34.64% | 31.35% | 39.36% | 43.18% | 36.94% +0.36% |
|       | AWQ | W4A8 | 34.86% | 29.69% | 40.98% | 41.27% | 36.57% |
|       | AWQ w/ IS | W4A8 | 34.13% | 30.19% | 40.40% | 41.52% | 36.36% -0.21% |
|       | Omniquant | W4A8 | 34.39% | 31.84% | 42.28% | 43.77% | 37.74% |
|       | Omniquant w/ IS | W4A8 | 33.65% | 31.05% | 40.17% | 43.18% | 36.72% -1.02% |
| LLaMA-2-13B | FP16 | W16A16 | 54.43% | 44.27% | 63.41% | 60.76% | 55.68% |
|       | GPTQ | W4A8 | 51.88% | 43.57% | 62.01% | 60.21% | 54.24% |
|       | GPTQ w/ IS | W4A8 | 52.18% | 43.27% | 61.33% | 60.83% | 54.27% +0.03% |
|       | AWQ | W4A8 | 50.07% | 41.75% | 60.90% | 59.19% | 52.76% |
|       | AWQ w/ IS | W4A8 | 49.65% | 42.64% | 59.80% | 58.45% | 52.40% -0.36% |
|       | Omniquant | W4A8 | 52.56% | 43.21% | 62.56% | 60.67% | 54.61% |
|       | Omniquant w/ IS | W4A8 | 52.05% | 43.14% | 61.72% | 60.02% | 54.09% -0.52% |
| LLaMA-2-70B | FP16 | W16A16 | 65.16% | 57.79% | 80.44% | 74.61% | 69.11% |
|       | GPTQ | W4A8 | 62.49% | 55.17% | 78.55% | 73.01% | 66.86% |
|       | GPTQ w/ IS | W4A8 | 62.42% | 55.14% | 78.39% | 72.73% | 66.74% -0.12% |
|       | AWQ | W4A8 | 63.44% | 55.86% | 78.45% | 72.12% | 67.11% |
|       | AWQ w/ IS | W4A8 | 63.70% | 55.33% | 78.00% | 71.75% | 66.89% -0.22% |
| Mixtral-8x7B | FP16 | W16A16 | 64.46% | 61.30% | 81.51% | 77.39% | 70.50% |
|       | GPTQ | W4A8 | 61.70% | 58.78% | 78.78% | 73.81% | 67.61% |
|       | GPTQ w/ IS | W4A8 | 61.66% | 57.55% | 77.58% | 73.60% | 67.02% |
|       | AWQ | W4A8 | 64.48% | 60.17% | 80.05% | 75.20% | 69.44% |
|       | AWQ w/ IS | W4A8 | 62.85% | 59.18% | 79.07% | 74.58% | 68.32% |
|       | Omniquant | W4A8 | 63.00% | 58.78% | 80.21% | 75.69% | 68.79% |
|       | Omniquant w/ IS | W4A8 | 62.17% | 58.81% | 79.92% | 75.17% | 68.34% |

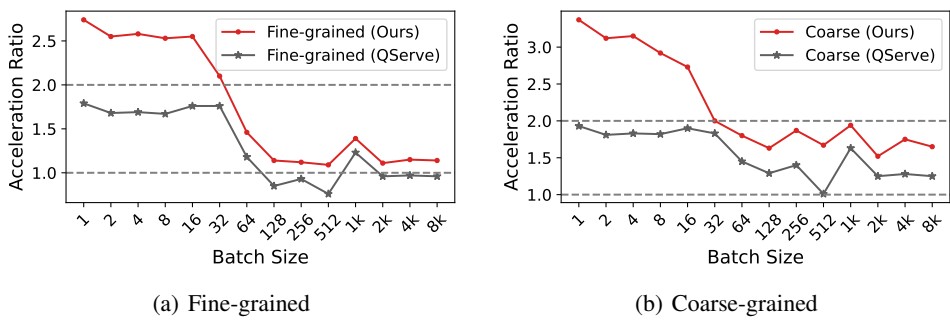

(a) Fine-grained      (b) Coarse-grained

Figure 7: Kernel (N=4096,K=4096) speed comparison with QServe. The acceleration ratio is against FP16.

amplifier of 1024. Both activation quantization is set per token. Note Integer Scale is robust on all model sizes, while VS-Quant fails on LLaMA-2-70B. While VS-Quant attempts to solve quantization problems for ResNet-50 and BERT, but fails to generalize to large models like LLaMA-2-70B (see Table (d)). It is a two-level quantization approach that involves clipping while we don't involve clipping search.

DGQ (Zhang et al., 2024) is based on VS-Quant which is a quantization scheme on fine-grained scales, leading to larger clipping and rounding errors. We are a different method. Note that DGQ

Table 10: Comparison our Integer Scale W4A8 with VSQuant's W4A8 with quantized scale on C4.

| Model | Quantization | Dataset | Group size | VS-Quant | Integer Scale |
|-------|--------------|---------|------------|----------|---------------|
| LLaMA-2-7B | W4A8 | C4 | 128 | 7.6122 | 7.5746 |
| LLaMA-2-13B | W4A8 | C4 | 128 | 6.6908 | 6.6849 |
| LLaMA-2-70B | W4A8 | C4 | 128 | NaN | 5.7814 |

doesn't achieve practical gain over W8A8 due to its inefficient design on dequantization, discussed also in section IV b) of QServe Lin et al. (2024).

### B.4 MAX ACTIVATION VALUES PER LAYER

To verify whether our amplifier choice is feasible and not causing overflows, we illustrate the maximum layerwise activation values on the investigated models in Figure 8. It appears no layer's output goes near the INT32 upper bound. We refrain from selecting a higher amplifier to improve performance since it will generate few gains and in the meantime increase the overflow risk.

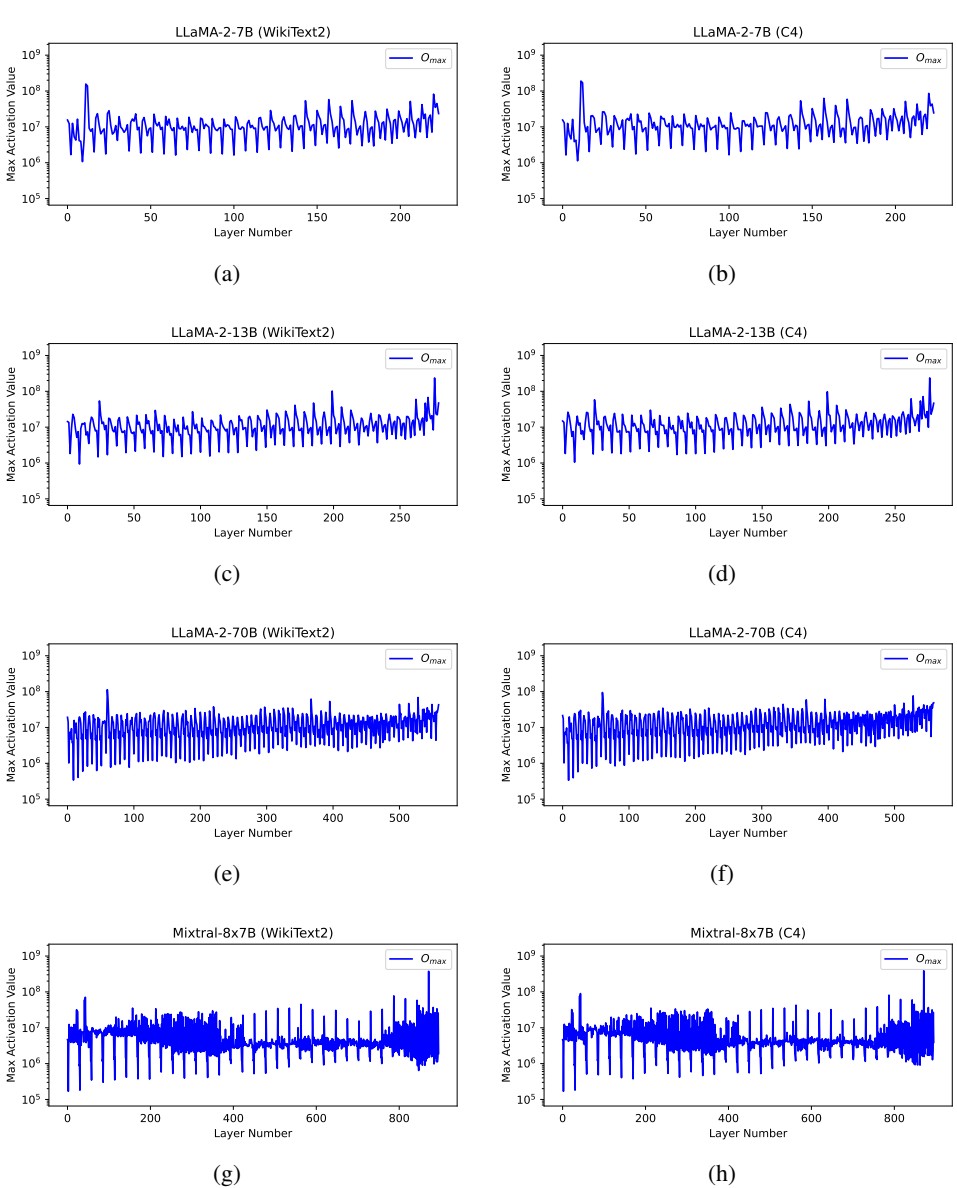

Figure 8: Maximum activation values per layer of quantized LLaMA-2 models and Mixtral 8x7B using an amplifier of 1024.

