# OpenReview forum: "Integer Scale: A Free Lunch for Faster Fine-grained Quantization of LLMs"
_ICLR.cc/2025/Conference — ICLR 2025 Conference Withdrawn Submission_

### Official Review · Reviewer_jSZC · 2024-10-17

**Soundness:** 3
**Presentation:** 3
**Contribution:** 2
**Rating:** 5
**Confidence:** 5

**Summary:**

In the realm of large-scale language models, weight-only quantization is a common approach for low-bit quantization. Typically, per-group fine-grained quantization is employed to minimize quantization errors. However, this fine-grained method isn't computationally efficient on GPUs because the results of grouped matrix multiplications require time-consuming type conversions from int32 to float32 (I32toF32).

To address this challenge, the paper identifies a key bottleneck: the per-group scales of weights are stored in float32 format, necessitating these costly type conversions after each grouped matrix multiplication. Building on this insight, the authors introduce the "Integer Scale" method. This technique quantizes the per-group weight scales into int32 integers, allowing the grouped matrix multiplication results to be directly multiplied by the integer scales without any type conversion. After computation, the overall result is scaled by dividing with the amplification factor.

This Integer Scale method acts as a plugin that can be seamlessly integrated into existing quantization frameworks like GPTQ and AWQ. By doing so, it effectively overcomes the drawbacks of per-group quantization. Extensive experiments conducted on the LLaMA series models demonstrate that using integer-scaled quantization maintains nearly the same precision as the original float scales while significantly enhancing inference speed.

**Strengths:**

a. Originality
The paper offers a reasonably original contribution by addressing the performance issues inherent in per-group quantization for low-bit weight-only quantization methods. By introducing an integer scaling method in this context, it directly and effectively resolves the problem where grouped matrix multiplication results require extensive type conversions (I32toF32), which previously diminished any speed advantages.

b. Quality
The paper demonstrates thorough experimentation. It validates the precision of the proposed method across a substantial portion of the LLaMA series models and combines it with several common weight-only methods such as GPTQ and AWQ. In terms of speed, it provides comparisons with W4A8 per-group float scale and W4A16, and also contrasts its approach with QServe, another method that tackles the same issue.

c. Clarity
The paper is well-written and easy to comprehend. It clearly articulates the problem at hand and presents a straightforward solution, making it accessible to readers without sacrificing technical depth.

d. Significance
The significance of the problem addressed is underscored not only in this paper but also in prior work like QServe, indicating that it is an urgent issue in the field. By providing an effective solution, the paper contributes valuable insights that are timely and relevant to ongoing advancements in large-scale model quantization.

**Weaknesses:**

a. Limitations of the Integer Scale Method and Potential for Overflow
The Integer Scale method proposed in the paper essentially involves converting floating-point scales to fixed-point integers. While this approach effectively reduces the need for type conversions (I32toF32) and improves computation speed in the cases presented, it may not be robust in scenarios where weight scales have larger values. If the float scales are significantly large, directly multiplying by the amplification factor 'm' could introduce precision issues. Moreover, there's a risk that the Integer Scale method could lead to int32 computation overflows when dealing with larger scales or different model architectures, potentially compromising the accuracy and reliability of the quantization.

b. Inconsistent Comparison with QServe
In the speed comparisons between the proposed method and QServe, the paper does not appear to maintain consistent conditions, which could affect the validity of the results. Specifically, QServe utilizes asymmetric quantization for weights, whereas the proposed method employs symmetric quantization. This discrepancy means that the comparison might not accurately reflect the inherent performance differences between the two methods. For a fair and meaningful evaluation, it would be necessary to control this variable by ensuring that both methods use symmetric quantization for weights during the comparison. Without this consistency, the claimed speed advantages of the Integer Scale method over QServe may not be entirely attributable to the method itself.

**Questions:**

a. Fair Comparison with QServe
Is it possible for the authors to control the experimental conditions to ensure a fair and consistent comparison with QServe? Specifically, aligning the quantization settings—such as using symmetric quantization for weights in both methods—would provide a more accurate assessment of the performance differences. A controlled comparison would help isolate the benefits of the proposed Integer Scale method and offer clearer insights into its advantages over QServe.

b. Applicability to Other Models Like the Qwen2 Series
Can the authors provide experimental results demonstrating the effectiveness of the Integer Scale method on other model architectures, such as the Qwen2 series? Expanding the evaluation to include these models would help establish the generalizability of the method across different types of networks.

c. Performance on Smaller Models
The experiments presented focus on models with 7B parameters and above. Have the authors conducted experiments on smaller models, such as those with 1.5B or 3B parameters? Providing results on smaller-scale models would offer insights into the scalability and effectiveness of the Integer Scale method across a broader range of model sizes.

---

### Official Review · Reviewer_9Yaj · 2024-11-03

**Soundness:** 1
**Presentation:** 1
**Contribution:** 1
**Rating:** 1
**Confidence:** 5

**Summary:**

This paper introduced Integer Scale, a novel post-training quantization approach designed to improve the inference speed of large language models without sacrificing accuracy. Integer Scale addresses the bottleneck in fine-grained quantization by eliminating the need for additional calibration or fine-tuning.

**Strengths:**

The method of this work was simple.

**Weaknesses:**

1. The proposed method in this paper is rather simplistic, essentially functioning as a straightforward parameter manipulation technique. The approach lacks the depth or complexity one might expect in advanced quantization methods for large language models.

2. The paper’s structure and presentation are also suboptimal, deviating from standard conventions in academic writing. This lack of clarity and cohesion detracts from its readability and scholarly rigor.

3. Furthermore, the comparative analysis is insufficient, and the reported experimental results reveal only marginal performance advantages. The limited scope of the improvements raises questions about the practical impact and innovation of the proposed method.

**Questions:**

The comparative analysis is insufficient, and the reported experimental results reveal only marginal performance advantages. The limited scope of the improvements raises questions about the practical impact and innovation of the proposed method.

---

### Official Review · Reviewer_B9W1 · 2024-11-03

**Soundness:** 3
**Presentation:** 3
**Contribution:** 3
**Rating:** 6
**Confidence:** 3

**Summary:**

This paper introduces Integer Scale, a post-training quantization (PTQ) method tailored for LLMs. Integer Scale seeks to overcome the efficiency limitations of LLM inference by employing fine-grained quantization while maintaining accuracy. It functions as a plug-in for established quantization methods such as GPTQ, AWQ, and Omniquant. Experimental results indicate that Integer Scale achieves performance enhancements of 2X for the Mixtral-8x7B model and 2.3X for the LLaMA-3 model compared to FP16 baselines.

**Strengths:**

Integer Scale addresses inference bottlenecks in fine-grained quantization methods while maintaining accuracy. Integer Scale achieves up to a 1.85x end-to-end speed boost compared to FP16 precision and outperforms existing methods such as W4A16 and W4A8, all while requiring no additional calibration or fine-tuning—making it a "free lunch" solution for implementation.

The comprehensive evaluation covers a variety of model architectures, including Llama models and the MoE model (Mixtral 8x7B). Integer Scale can be seamlessly integrated as a plug-and-play addition to most fine-grained quantization techniques, including GPTQ, AWQ, and OmniQuant. By resolving the speed limitations of fine-grained quantization, this method enhances the practicality of deploying large language models effectively without compromising performance.

**Weaknesses:**

The Integer Scale formulation appears to build on concepts introduced in VSQuant, which suggested the use of FP16 scaling factors for model quantization, followed by quantizing these factors to integers. Additionally, the implementation seems to draw inspiration from established frameworks such as FastGEMM and Atom. This overlap may make it somewhat challenging to distinguish the original contributions of this paper.

**Questions:**

Please address the questions raised in the weaknesses section.

---

### Official Review · Reviewer_jL9D · 2024-11-04

**Soundness:** 2
**Presentation:** 3
**Contribution:** 1
**Rating:** 3
**Confidence:** 4

**Summary:**

This paper proposes a calibration-free method called Integer Scale for quantizing floating-point scales into integers, which can be utilized in symmetric per-group quantization methods. The Integer Scale approach results in a significant speed improvement compared to original floating-point quantization methods.

**Strengths:**

The article is well-written and includes numerous diagrams, enhancing its clarity and facilitating understanding. The proposed method is simple and straightforward. Extensive experiments have been conducted, comparing both accuracy and speed against various alternative methods.

**Weaknesses:**

This paper lacks novelty, as the concept of Integer Scale has been discussed in numerous quantization studies[1,2,3]. The absence of speed and accuracy comparisons with previous full integer quantization methods renders the findings unconvincing. Additionally, the proposed method of using a scale amplifier to convert floating-point scale to integer scale aligns with the widely used round-to-nearest fixed-point quantization technique.

[1] Training High-Performance and Large-Scale Deep Neural Networks with Full 8-bit Integers

[2] Integer Quantization for Deep Learning Inference: Principles and Empirical Evaluation

[3] I-LLM: Efficient Integer-Only Inference for Fully-Quantized Low-Bit Large Language Models

**Questions:**

- This paper exclusively conducts experiments on symmetric quantization and does not address asymmetric quantization methods. While precision comparisons are limited to symmetric methods, latency is compared with asymmetric methods, such as Qserve, which is not a fair comparison. I recommend that the authors include precision and speed comparisons with the same algorithm under consistent settings.
- Integer scaling is not a novel research direction, as it has been explored in numerous prior studies[1,2]. This article lacks a comparison with full integer models regarding method, accuracy and speed.
- The authors state in the introduction that the proposed method “we have achieved a new Pareto-front of speed vs. accuracy”; however, this claim is not demonstrated in the subsequent content.

[1] Integer Quantization for Deep Learning Inference: Principles and Empirical Evaluation

[2] I-LLM: Efficient Integer-Only Inference for Fully-Quantized Low-Bit Large Language Models

---

### Note · Authors · 2024-11-25

**Comment:**

We thank every human reviewer for the concrete feedback. We are looking forward to including our latest results to strengthen our paper for the next venue.

We are suspicious that Reviewer 9Yaj might have used an AI-generated review (with no citation or proof to back up his/her viewpoints), sabotaging the top conference's prestige. We recommend that these reviewers be banned from the platform.

Sincerely,
The Authors

**Withdrawal Confirmation:**

I have read and agree with the venue's withdrawal policy on behalf of myself and my co-authors.